# Vaccination against the Protozoan Parasite *Histomonas meleagridis* Primes the Activation of Toll-like Receptors in Turkeys and Chickens Determined by a Set of Newly Developed Multiplex RT-qPCRs

**DOI:** 10.3390/vaccines9090960

**Published:** 2021-08-27

**Authors:** Taniya Mitra, Beatrice Bramberger, Ivana Bilic, Michael Hess, Dieter Liebhart

**Affiliations:** 1Clinic for Poultry and Fish Medicine, Department for Farm Animals and Veterinary Public Health, University of Veterinary Medicine, 1210 Vienna, Austria; 1245157@students.vetmeduni.ac.at (B.B.); ivana.bilic@vetmeduni.ac.at (I.B.); michael.hess@vetmeduni.ac.at (M.H.); dieter.liebhart@vetmeduni.ac.at (D.L.); 2Christian Doppler Laboratory for Innovative Poultry Vaccines (IPOV), University of Veterinary Medicine, 1210 Vienna, Austria

**Keywords:** pro-inflammatory cytokines, extracellular pathogen, innate immune response, poultry, blackhead disease, histomonosis

## Abstract

Histomonosis in turkeys and chickens is caused by the extracellular parasite *Histomonas meleagridis*, but the outcome of the disease varies depending on the host species. So far, studies on the immune response against histomonosis focus mainly on different traits of the adaptive immune system. Activation of toll like receptors (TLR) leads to the interplay between cells of innate and adaptive immunity with consequences on B and T cell clonal expansion. Therefore, the present investigation focused on the interaction of virulent and/or attenuated histomonads with the innate immune system of turkeys and chickens at 4, 10, 21 days post inoculation. The expression of TLRs (*TLR1A*, *1B*, *2A*, *2B*, *3*, *4*, *5*, *6*(Tu), *7*, *13*(Tu) and *21*(Ch)) and pro-inflammatory cytokines (*IL1β* and *IL6*) were analysed in caecum and spleen samples by RT-qPCR. Most frequent significant changes in expression levels of TLRs were observed in the caecum following infection with virulent parasites, an effect noticed to a lower degree in tissue samples from birds vaccinated with attenuated parasites. *TLR1B*, *2B* and *4* showed a continuous up-regulation in the caecum of both species during infection or vaccination, followed by challenge with virulent parasites. Vaccinated birds of both species showed a significant earlier change in TLR expression following challenge than birds kept non-vaccinated but challenged. Expression of TLRs and pro-inflammatory cytokines were associated with severe inflammation of diseased birds in the local organ caecum. In the spleen, changes in TLRs and pro-inflammatory cytokines were less prominent and mainly observed in turkey samples. In conclusion, a detailed comparison of TLRs and pro-inflammatory cytokines of the innate immune system following inoculation with attenuated and/or virulent *H. meleagridis* of two avian host species provides an insight into regulative mechanisms of TLRs in the development of protection and limitation of the disease.

## 1. Introduction

*Histomonas meleagridis* is a widespread flagellated parasite of poultry which causes histomonosis (syn. blackhead disease, histomoniasis or infectious typhlohepatitis) [1]. In turkeys, histomonosis can be fatal, resulting in typhlohepatitis with severe pathological lesions characterized by an acute inflammation of caecum and liver. In chickens, the disease is less destructive, but can cause similar lesions in both organs, with consequences on performance [2]. Histomonads primarily target the caecum, before reaching the liver via the portal vein [3]. Today the disease is re-emerging due to the ban of prophylactic and therapeutic drugs [2]. Experimental vaccination with in-vitro attenuated parasites was shown to be effective in turkeys and chickens against a challenge and might be used in the future to protect birds [4]. However, knowledge of the immune response of poultry against the parasite is limited and mainly focused on the adaptive immunity [5]. So far, no studies have been performed investigating the innate immune response against *H. meleagridis*, especially toll-like receptors (TLRs), together with pro-inflammatory cytokines expression.

TLRs are innate immune pattern recognition receptors (PRRs) that are of importance for defending the host from pathogens and to keep up immune homeostasis [6]. TLRs sense the presence of conserved microbial structures in the environment and play a significant role in inflammation, immune cell regulation, survival and proliferation. The discovery and characterization of the TLR family in chickens and turkeys has incited new interest in the field of innate immunity [7]. It is common knowledge that these receptors have a vital role in microbial recognition, induction of antimicrobial genes and the control of adaptive immune responses. The chicken is predicted to have two TLR2 isoforms (TLR2 types 1 and 2), two TLR1/6/10 orthologs, and a single TLR3, TLR4, TLR5, and TLR7. In addition, chickens have two TLRs that appear absent in the mammalian species, namely TLR13 and TLR21 [8]. In turkeys, so far TLR1A, TLR2A, TLR1B, TLR2B, TLR3, TLR4, TLR5, TLR6 and TLR13 have been identified only by in-silico analyses [9]. Very few studies investigated TLR responses against intracellular parasites in chickens, specifically only against *Eimeria* species [10,11,12,13,14]. *TLR1A*, *4*, *5*, *7* and *21* were reported to be up-regulated in caecum after *Eimeria tenella* infection [11]. In the spleen, *TLR3* and *15* were the main relevant TLRs [13]. Taken together, the expression patterns of TLRs during innate immune response against an extracellular protozoan parasite in poultry have not been investigated up to now. The hypothesis of the current study is that TLR signaling can function as a decisive determinant of the innate immune and inflammatory responses in local and systemic organs following vaccination and/or infection with *H. meleagridis*. Furthermore, pro-inflammatory cytokines were analysed in the same tissues to monitor a relation to relevant TLRs. For that purpose, RT-qPCR assays assessing the expression of TLR and cytokines in caecum and spleen of chickens and turkeys were established. Consequently, differences in the activation patterns of TLRs and pro-inflammatory cytokines in turkeys and chickens were revealed after vaccination and/or infection with attenuated respectively virulent *H. meleagridis* locally and systematically.

## 2. Materials and Methods

### 2.1. Animal Trial

The animal trial was carried out as previously described [15]. Briefly, 60 turkeys (B.U.T. 6; Aviagen Turkeys Ltd., Tattenhall, UK) and 60 specific pathogen free (SPF) layer chickens (VALO, BioMedia, GmBH, Osterholz-Scharmbeck, Germany) were used in the present work. The birds were individually marked at their first day of life using tags that were subcutaneously attached. The animal trial was approved by the institutional ethics committee, ethics committee of the University of Veterinary Medicine, Vienna, Austria and the national authority according to §26 of the Law for Animal Experiments, Tierversuchsgesetz—TVG (license number bmwf GZ 68.205/0147-II/3b/2013).

### 2.2. Preparations of Parasites for Inoculation

The clonal culture of *H. meleagridis*/Turkey/Austria/2922-C6/04 [16] was selected to vaccinate and infect the birds. For vaccination, in vitro attenuated histomonads (passage 295) were used and for infection virulent, short-term cultured histomonads (21 passages) were applied, as previously mentioned [17]. The cultures were kept at −150 °C until inoculation. 6 × 10^5^ cells of histomonads in 600 µL culture medium containing Medium 199 with Earle’s salts, L-glutamine, 25 mM HEPES and L-amino acids (Gibco™ Invitrogen, Lofer, Austria), 15% foetal calf serum (FCS) (Gibco™ Invitrogen) and 0.66 mg rice starch (Sigma-Aldrich, Vienna, Austria) were administered per bird. Equal amounts of the inoculum were given orally and cloacally using a syringe together with a crop tube or a pipette, respectively. Birds of the control groups were inoculated with same volume of pure culture medium.

### 2.3. Trial Setup

Unmedicated turkey starter feed (Vitakorn, Pöttelsdorf, Austria) or chicken starter feed (Likra Tierernährung GmbH, Linz, Austria) were provided ad libitum. The feed was removed for 5 h directly after inoculation. For each species 15 birds/group were kept in separate rooms depending on the inoculum: vaccinated turkeys (VT), vaccinated chickens (VC), infected turkeys (IT), infected chickens (IC), vaccinated and infected turkeys (VIT), vaccinated and infected chickens (VIC), control turkeys (CT) and control chickens (CC). Vaccination with attenuated live histomonads of groups VIT and VIC was administered on the day of hatch. At the age of 28 days, IT, IC, VIT and VIT were infected with the virulent strain. On the same day, VT and VC were vaccinated with attenuated strain of histomonads and control birds (CT and CC) were inoculated with culture medium. Accordingly, three previously determined birds (ascending order of tag numbers) per group were killed at 4, 7, 10, 14 and 21 days post inoculation (DPI). Throughout the experiment, clinical signs such as ruffled feather, diarrhea and depression was monitored in every group. The lesion score for caecum and liver was determined during the necropsy according to Windisch and Hess, 2010 and Zahoor et al., 2011 [18,19]. The samples were stored at −80 °C in RNA*later*^®^ (Qiagen, Hilden, Germany) [15]. In the present study, samples of spleen and caecum collected at 4, 10 and 21 DPI were used to investigate the expression of TLRs and pro-inflammatory cytokines at an early, intermediate and late time point after vaccination, respectively infection. A schematic diagram is shown in Figure 1.

### 2.4. Gene Selection

All the TLRs so far identified for turkeys and chickens were included in the study: *TLR1A*, *1B*, *2A*, *2B*, *3*, *4*, *5*, *6*(Tu), *7*, *13*(Tu) and *21*(Ch) [8,9,20] from both poultry species (Table 1). Along with the TLRs, pro-inflammatory cytokines were selected for their expression. Hence, *IL1β* and *IL6* were investigated to identify changes of further immune markers beside TLRs that have been already described to play a crucial role during histomonosis [21].

### 2.5. Total RNA Extraction and Analysis for Purity and Integrity

RNA extraction was performed as previously described [22]. Total RNA was prepared from tissue samples of spleen and caecum that were stabilized in RNA*later*^®^ (Qiagen, Hilden, Germany) and stored at −80 °C. The samples were separately homogenized using QIAshredders (Qiagen, Germany) before total RNA was extracted by RNeasy^®^ mini kit (Qiagen, Germany) according to manufacturer’s instructions. The RNA quality of all the tissue samples was determined by NanoDrop 2000 (ThermoFisher scientific, Vienna, Austria) to assure that the RNA was free from contaminates. The nucleic acid purity was analysed with A260/280 ratio ranging from 1.5 to 2.3 and samples with an equal or above 2 ratio of A260/230 were selected for further use. The RNA quality and quantity was further surveyed using Bioanalyzer 2100 (Agilent Technologies, Waldbronn, Germany). Thereby, the RNA concentration and the integrity as well as the presence or absence of degradation products were ascertained by identifying the entire electrophoretic trace of every sample. Samples with an RNA integrity number (RIN) from 6.5 to 9.5 were considered to be used for RT-qPCR (Appendix A).

### 2.6. RT-qPCR

Primer and probes targeting a highly conserved region of each TLR gene were designed for chicken and turkey samples according to NCBI database (www.ncbi.nlm.nih.gov) (Table 1) and GenScript real-time PCR (TaqMan) primer design software (Genscript Biotech, Piscataway, NJ, USA) (www.genscript.com) with default settings. NCBI database and GenScript software were accessed on 4 December 2017 for *TLR1A*, *1B*, *2A*, *2B*, *3* (Ch), *4*, *5*, *6* (Tu). The same database and software were used for *TLR 3* (Tu), *7*, *13* (Tu), *21* (Ch) on 29 November 2019. All the target genes were first standardized for singleplex RT-qPCRs with SYBR green reagent (Agilent Technologies, Germany). The size of the final product was verified by agarose gel electrophoresis. Afterwards, the detection of genes was performed by multiplex RT-qPCRs. Along with slope, R^2^, the efficiency difference between singleplex and multiplex was kept at the same level (below 5% variation). Details about the multiplex setup is given in the Appendix A. The primer and probe sequences targeting pro-inflammatory cytokines were reported previously by Powell et al. [21].

For one-step RT-qPCR, TaqMan chemistry and Brilliant III Ultra-Fast QRT-PCR master mix kit were applied (Agilent Technologies, Germany). For the RT-qPCR reaction, the AriaMx real-time PCR system (Agilent Technologies, Germany) with the Agilent AriaMx v1.7 software (Agilent Technologies, Germany) was used. The following thermal cycle profile was applied: 1 cycle of reverse transcription at 50 °C for 10 min followed by 95 °C for 3 min of hot start, 40 cycles of amplification at 95 °C for 5 s and 60 °C for 10 s. The primers were used in concentrations from 200 nM to 900 nM and the probes with 100 nM. The samples were run in duplicate and NRT (non-reverse transcriptase) as well as NTC (non-template control) were performed to exclude contamination. For the gene expression analysis, the mean CT value of each duplicate was used. The RT-qPCR investigations have been performed according to the MIQE guidelines [23]. To account for the variation in sampling and RNA preparation, the CT values for all genes were normalized using CT values of previously reported reference genes *RPL13* and *TFRC* [22]. To evaluate the results, all the values were given as fold change by using 2^−∆∆CT^ formula [24]. In this formula ΔCT was calculated for each group separately, where ΔCT = CT (a target gene)—CT (a reference gene). Followed by ΔΔCT = ΔCT (a treated sample) − ΔCT (a control sample), final 2^−∆∆CT^ to get fold change values.

### 2.7. Statistical Analysis

For lesion scores, mean lesion scores of caecum and liver were calculated for every group of both species. Total lesion score of all the birds from a specific day in a group were summed and divided by the total number of the birds in the respective group at that specific time point.

A preliminary assessment on normal distribution assumptions was carried out at first. Every individual RT-qPCR dataset was firstly verified by Shapiro-Wilk test with histogram and Q-Q plots. Afterwards, the mean values from vaccinated and/or infected groups were compared with the control group of the respective species with an unpaired student’s *t*-test that was used. A *p*-value less than 0.05 (* *p* ≤ 0.05) was considered to be statistically significant. For graphical representation of data, vaccinated and/or infected groups were compared with the control group of the respective species and given as fold change value.

## 3. Results

### 3.1. Establishment of RT-qPCR

The singleplex RT-qPCR was evaluated for the specificity of the product by melting curve analysis. Afterwards, multiplex RT-qPCR was established following the MIQE guidelines [23]. The information on the established multiplex RT-qPCR is given in Appendix A and the mean RIN value of all the samples are given in Appendix A.

### 3.2. Clinical Signs and Lesion Scores

Non-vaccinated but infected turkeys (group IT) showed first clinical signs, such as depression, diarrhoea and ruffled feathers, starting on 7 DPI. Due to severity of histomonosis all birds of this group had to be euthanized before 14 DPI. In contrast, no clinical signs were noticed in any other group of turkeys and chickens. Different grades of pathological changes and the mean LS on the respective sampling day of every group determined during the post mortem procedure are shown in detail in the previously published work [15]. A brief description of mean LS in caecum and liver of both species at 4, 10 and 21 DPI is given in Table 2. Due to the fatality of the disease in infected turkeys no data is available from this species at 21 DPI. None of the control birds showed lesions at any sampling day.

### 3.3. Gene Expression of TLRs

#### 3.3.1. Local TLR Expression in the Caecum

In the caecum of vaccinated turkeys, *TLR1B* was significantly up-regulated at 4 DPI and *TLR7* was significantly down-regulated at 10 DPI (Figure 2) compared to control group (CT). In VC, *TLR4* and *TLR5* were significantly downregulated at 10 DPI, and for the latter one at 21 DPI as well (Figure 2). In both bird species, there was no significant up-regulation of any *TLR*, except for *TLR1B* in VT.

In VIT, *TLR1B* was significantly up-regulated at 10 DPI, while this was the case for *TLR2B* at 4 DPI. *TLR4* was up-regulated at 10 and 21 DPI, *TLR6* at 21 DPI and *TLR7* at 10 DPI in the same organ. In VIC, *TLR1B*, *TLR2B* and *TLR21* were significantly up-regulated at 4 DPI. *TLR4* and *5* displayed significant down-regulation at 10 or 21 DPI, respectively. All turkeys that received vaccination and infection as mentioned above showed consequences on upregulation of *TLR1B*, *TLR2B*, *TLR4*, *TLR6* and *TLR7*, whereas in chickens of the equivalent group, *TLR1B*, *TLR2B* and *TLR21* were found to be increased in contrast to *TLR4* and *TLR7* which were decreased.

In the caecum of IT group, *TLR1B*, *2B* and *4* were significantly up-regulated on days 4 and 10 after infection. At the same time, *TLR3* was downregulated, with a significant decrease observed at 10 DPI. In the caecum of infected chickens, *TLR1A*, exhibited a significant up-regulation at the last measured time point at 21 DPI. In the same group of birds, *TLR1B* and *2B* showed significant up-regulation at 10 DPI. *TLR3* increased at 4 DPI but decreased significantly at 10 DPI. *TLR4* and *5* were significantly downregulated at 4 and 21 DPI, respectively. Altogether, *TLR1A*, *2B*, *3*, *4* in IT and *TLR1A*, *1B*, *2B*, *3*, *4* and *5* in IC were the relevant TLRs that showed alterations in the primary infected organ caecum.

An overview of significant changes in the expression of TLRs is given in Table 3.

#### 3.3.2. Systemic TLR Expression in the Spleen

In the spleen of vaccinated turkeys, *TLR3* showed a significant up-regulation at 10 DPI (Figure 3). *TLR5* and *6* were significantly downregulated at 10 and 4 DPI, respectively. In VC, no significant changes were observed in the spleen (Figure 3).

In VIT and VIC, no significant changes were observed.

In the spleen of IT, *TLR3* was significantly up-regulated at 10 DPI, but at the same time *TLR5* was significantly down-regulated. No significant changes were observed in IC. An overview of significant changes in the expression of TLRs has been summarized in Table 3.

### 3.4. Gene Expression of Pro-Inflammatory Cytokines

#### 3.4.1. Caecum

Vaccinated birds (VT and VC) showed no significant changes in caecum of both cytokines. *IL1β* was significantly up-regulated at 4 DPI in VIT. VIT and VIC exhibit significant up-regulation at 4 DPI for IL6, which was continued in VIT at 10 DPI. In both groups IT and IC, with birds being only infected but non-vaccinated, significant up-regulation for pro-inflammatory cytokine *IL1β* at 4 DPI was noticed, an effect additionally seen at 10 DPI in IC. *IL6* was significantly up-regulated at 4 and 10 DPI for IT and only at 4 DPI for IC (Figure 4).

#### 3.4.2. Spleen

Vaccinated and vaccinated plus infected groups showed no significant changes at any time points. *IL6* was significantly up-regulated in infected turkeys and chickens at 4 DPI and additionally at 10 DPI in IT (Figure 4). At the remaining time points, results did not vary significantly.

## 4. Discussion

In this study, we established a probe-based system to evaluate expression of TLRs from two poultry species that ensures a high specificity against the targeted genes. The established RT-qPCRs were accordingly applied to investigate TLRs expression in respect to vaccination and/or infection against an extracellular parasite in turkeys and chickens. Studies on TLR expressions in chickens against parasites are lacking, with the exception of intracellular parasite *Eimeria* sp. [10,11,12,13,14]. Furthermore, so far nothing is reported on turkeys’ toll like receptor expression profiles in regard to infection.

In the present work, except *TLR1B* in VT, only vaccinated compared to control birds of both species showed no significant changes of all investigated TLRs. In contrast, infected birds showed frequent changes in TLRs when comparing with control birds. This is in agreement with findings from the cellular immune response that clearly varies according to the virulence of *H. meleagridis* [15]. In the previously mentioned work, attenuated histomonads caused only few changes in different lymphocyte subpopulations and macrophages/monocytes as compared to the infection with the virulent strain, arguing for less immunopathogenic effects. Here, it could be shown that this process is already initiated in the innate immune response.

Vaccinated and infected turkeys showed more significant changes than chickens with the same treatment. The exclusive up-regulation of *TLR6* and *7* in the caecum of VIT and *TLR21* in VIC argues for a regulative effect against inflammation in protected birds following infection. It has been already reported previously that *TLR6* and *7* can play an important role in preventing aberrant immune responses in the intestine and lung tissues in mice, rats and guinea-pigs [25,26]. Furthermore, *E. tenella* infection in chicken up-regulates *TLR7* and *21* [11].

Highest variations in the TLRs expression were observed in the caecum of turkeys and chickens infected with virulent *H. meleagridis* without previous vaccination. This revealed a distinct response of the innate immune system against the virulent parasites. The expression of those TLRs can be related to the development of inflammation in the caecum, which was also found to be most severe at 4 and 10 DPI in both bird species. However, it has to be mentioned that the sampling of IT was not possible at later time point (21 DPI) due to the fatal outcome of histomonosis. Mean lesion score in chickens declined at 21 dpi, which was accompanied with lower variations in the expression of TLRs and cytokines at this time point. It can be speculated that in infected birds the up-regulation of the TLRs at 4 and 10 DPI is involved in activation of inflammation. The increased expression of the *TLR1A, 1B, 2B, 3* and *4* following infection with *H. meleagridis* is in certain agreement with the outcome of other parasitic infection studies in chickens, but also in humans [11,13,27,28,29]. TLR1A and TLR2B can form a heterodimer and control agonist-driven immune response by activating NF-κB (nuclear factor kappa-light-chain-enhancer of activated B cells). TLR2 on human macrophages and NK cells elicited leishmanicidal reactions via the release of different mediators like TNF-α, IFN-γ, nitric oxide (NO) and reactive oxygen species (Th1 response) [25,26]. It has been reported that stimulation of TLR2 causes down-regulation of TLR5 in human monocytes [30], which is in agreement with the down-regulation of *TLR5* in the caecum of IC in the present work. In turkeys, we observed a similar tendency in the spleen, but not in the caecum. This might indicate the uncontrolled activation of the immune system as previously described based on the expression of cytokines [20]. Following infection of chickens with *E. tenella*, *TLR3* and *15* were also up-regulated in the intestine and spleen beside *TLR1A*, *4*, *5*, *7* and 21 in caecum [11,13]. *TLR4* was shown to induce the expression of pro-inflammatory cytokines in humans [29]. In our study, *TLR4* was significantly up-regulated in IT and significantly down-regulated in IC, which went along with the observed high lesion score in the organs of infected turkeys compared to chickens.

In the spleen of turkeys which were only vaccinated, *TLR3* showed a significant up-regulation at 10 DPI with *TLR5* and *6* being significantly down-regulated at 10 and 4 DPI, respectively. In comparison, no significant changes were observed in the spleen of chickens which were only vaccinated.

Furthermore, the absence of significant changes in the spleen of VIT and VIC groups of birds indicated a controlled immune cascade which enabled the regulation of the host response. This is reflected in clinical and pathological findings of these birds, which did not show any signs of histomonosis and had lower lesion scores compared to the only infected group.

In the spleen of infected turkeys, *TLR3* was significantly up-regulated at 10 DPI, but at the same time *TLR5* was significantly down-regulated. No significant changes were observed in IC. The involvement of the systemic immune response against virulent histomonads by activation of *TLR3* in turkeys showed a higher reactivity as compared to chickens. A similar observation was also described by investigating the response of histomonad-specific lymphocytes [31]. This clearly shows that both vaccination and infection caused much lower systemic than local immune response.

The non-protected host species responded differently to infection with *H. meleagridis*. The expression of TLRs was marked by stronger alterations in local and systemic level in turkeys than in chickens similar to differences in the clinical and pathological outcome in these two bird species. However, in both species the local TLR expression was clearly increased, whereas the systemic immune response had a minor alteration. It is well known that in the acute stage of local inflammation different TLRs related with inflammation—such as *TLR2* and *4*—are expressed [32]. This was shown to be caused by host cells with direct contact to the pathogens as described earlier [33].

In regard to pro-inflammatory cytokine expressions, virulent histomonads caused more frequent significant up-regulation of *IL-1β* and *IL-6* in caecum of non-vaccinated birds as compared to the other groups. Previously, a significant up-regulation of IL-1β, *CXCLi2* and *IL-6* mRNA was observed in the caecal tonsils in chickens at early time point following *H. meleagridis* infection; however, the same effect could not be detected in turkeys [21]. Here, we found a significant expression of *IL-1β* and *IL-6* in caecal tissue of both species at an early stage. However, the divergence in these findings is most probably related to the variable cellular composition of caecal tonsil and caecum, which might be the reason for the different immune response to the same pathogen. Differences in the expression of the cytokines in the caecum as compared to the spleen correlated with the presence or absence of the parasite as observed for TLR variations. A direct dependence of pro-inflammatory cytokines and TLRs was described to be caused by several adaptor molecules, such as IRAK molecules, which are recruited to initiate transcription factor activation and subsequently induce cytokine and chemokine production [34].

In general, the expression of TLRs and pro-inflammatory cytokines was more prominent in turkeys, which was in coherence with the clinical outcome of the disease and postmortem lesions in these birds. TLR-induced pro-inflammatory response is crucial for the elimination of the invading pathogens, but uncontrolled immune activation leads to tissue damage [35]. TLRs do not exclusively act as a key player in the inflammatory process by promoting the production of inflammatory molecules but are crucial as regulatory contributors. This was apparent in the present work by the generally lower expression of pro-inflammatory cytokines in vaccinated birds. However, the factors that influence TLR induction of either pro-inflammatory or anti-inflammatory mediators are still to be elucidated. Further understanding of parasite derived TLR ligands would lead to new therapeutic and prophylactic strategies for parasitic infections. This is especially of interest as recombinant TLRs have been proven to have the potential to act as an adjuvant applied to control mammalian infections [36].

## 5. Conclusions

In conclusion, the present work revealed for the first time the innate immune response in poultry by measuring TLR expression against *H. meleagridis*, an extracellular protozoan parasite. An infection with virulent parasites showed a more intense innate immune response as compared to vaccination with attenuated *H. meleagridis*. Moreover, the present data confirm that the timely expression of TLRs might contribute towards the immune protection of turkeys and chickens against *H. meleagridis*. The lower up-regulation of TLRs indicated the absence of excessive immune response in infected chickens, reflecting their recovery compared to infected turkeys. Vaccination abrogated an increased TLR response, which alleviated inflammation in the tissue of birds subsequently infected with virulent *H. meleagridis* and its potential immunopathogenic effects. The expression of TLRs on mRNA level limits further conclusions. Therefore, in future investigations the abundance of TLR proteins should be elucidated. Moreover, interaction of specific immune cells with antigens of *H. meleagridis* would be compelling to investigate TLRs function on cell-pathogen interaction.

## Figures and Tables

**Figure 1 vaccines-09-00960-f001:**
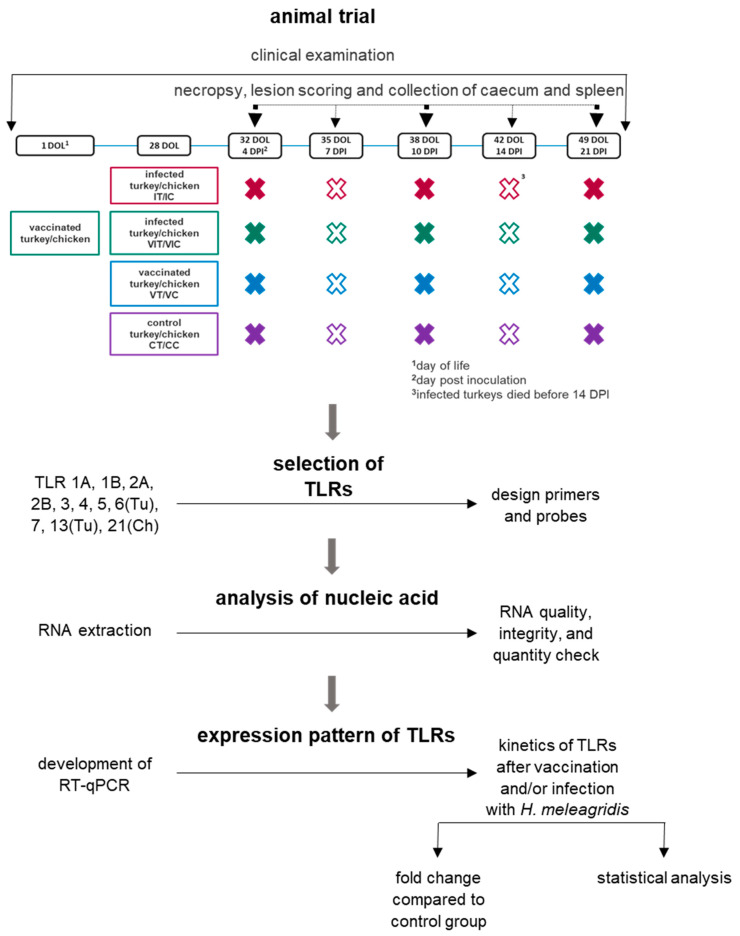
Schematic representation of the experimental setup and methods.

**Figure 2 vaccines-09-00960-f002:**
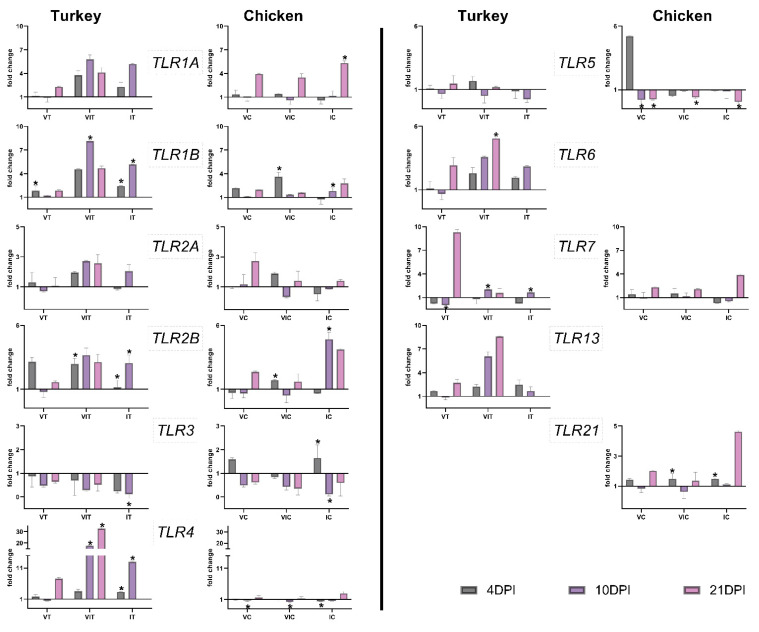
Expression pattern of TLRs in caecum of turkeys and chickens at 4, 10 and 21 days post inoculation (DPI) in vaccinated (VT/VC), vaccinated and infected (VIT/VIC) or infected (IT/IC) groups. Statistical differences were calculated for vaccinated and/or infected groups in comparison to the respective control group of the species (CT/CC). Significant changes are indicated as * (*p* value ≤ 0.05). Due to fatality of the disease, there were no birds left in group IT from 14 DPI onward.

**Figure 3 vaccines-09-00960-f003:**
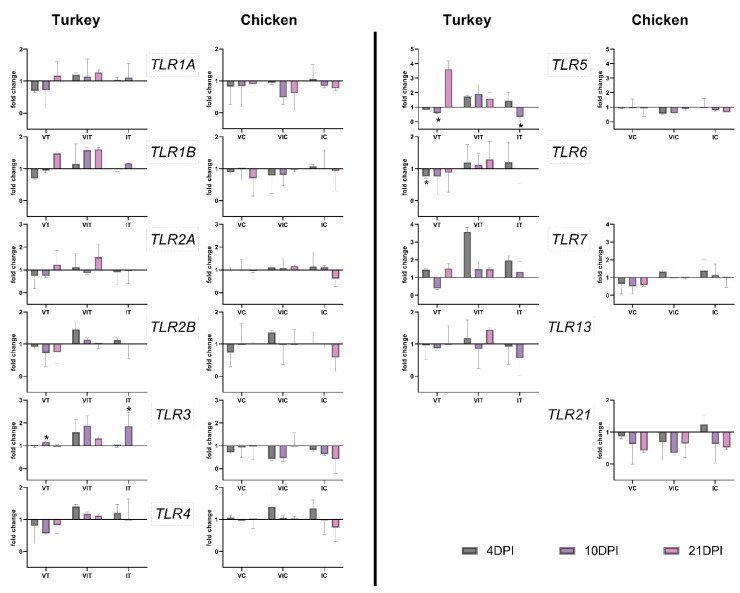
Expression pattern of TLRs in spleen of turkeys and chickens at 4, 10 and 21 days post inoculation (DPI) in vaccinated (VT/VC), vaccinated and infected (VIT/VIC) or infected (IT/IC) groups. Statistical differences were calculated for vaccinated and/or infected groups in comparison to the respective control group of the species (CT/CC). Significant changes are indicated as * (*p* value ≤ 0.05). Due to fatality of the disease, there were no birds left in group IT from 14 DPI onward.

**Figure 4 vaccines-09-00960-f004:**
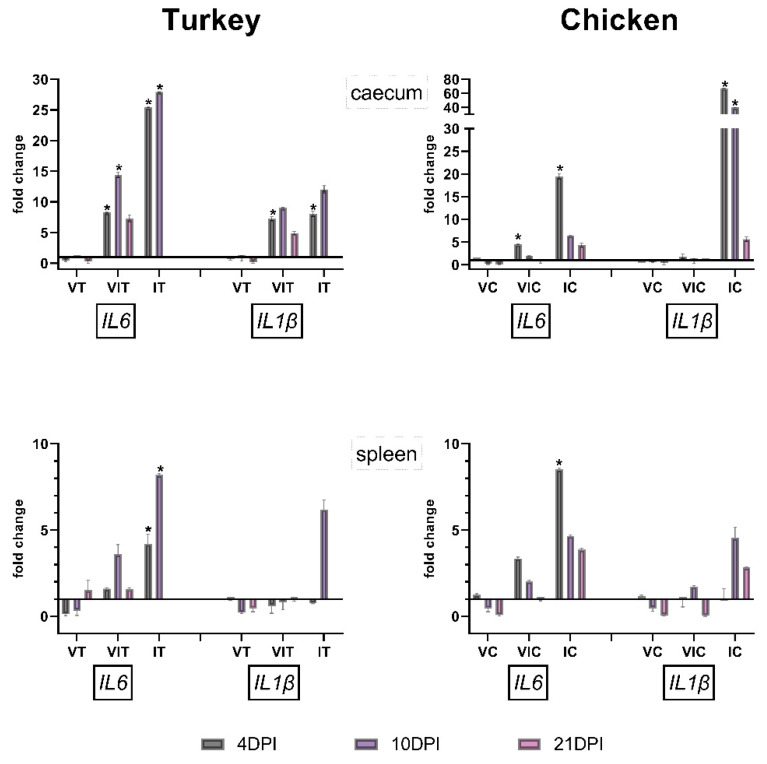
Expression pattern of pro-inflammatory cytokines in turkeys and chickens at 4, 10 and 21 DPI in only vaccinated (VT/VC), vaccinated and infected (VIT/VIC) or infected (IT/IC) groups in caecum and spleen. The statistical significance is indicated as * (*p* value ≤ 0.05). Significance differences were calculated for vaccinated and/or infected groups in comparison to the respective control group of the species (CT/CC). Due to severity of the disease, there were no surviving birds left in the group IT from 14 DPI onward.

**Table 1 vaccines-09-00960-t001:** Primers and probe sequences for the targeted genes.

*Gene*	Species	NCBI Accession No	Primers and Probe Sequences (5′–3′)
*TLR1A*	turkey	FJ477857.1	F: TGTCACTACGAGCTGTACTTTG R: CTCGCAGGGATAACATATGGAG P: FAM-TAGTCCTGATCTTGCTGGAGCCGA-TAMRA
	chicken	NM_001007488.4	
*TLR1B*	turkey	FJ477858.1	F: CCATCACAAGTTGTTTAGC R: TCCAGGTAGGTTCTCTTG P: HEX-CCTGATCTTGCTGGAGCCGA-BHQ1
	chicken	DQ518918.1	
*TLR2A*	turkey	FJ477860.1	F: CTGGCCCACAACAGGATAAA R: CCTCGTCTATGGAGCTGATTTG P: HEX-ACATGATCTGCAGCAGGCTGTGAA-BHQ1
	chicken	AB050005.2	
*TLR2B*	turkey	FJ477861.1	F: GATCCCCAAGAGGTTCTG R: CTGCTGTTGCTCTTCATC P: FAM-CTGCGGAAGATAATGAACACCAAGAC-BHQ1
	chicken	AB046533.2	
*TLR3*	turkey	XM_003205774.4	F: GCATAAGAAGGAGCAGGAAGA R: GGAGTCTCGACTTTGCTCAATA P: ROX-TGGTGCAGGAGGTTTAAGGTGCAT-BHQ2
	chicken	EF137861.1	
*TLR4*	turkey	XM_003211211.3	F: CATACAAGCCACTCCAAGCC R: AGGATTTCCAGGGCTGAGTC P: CY5-CACAGCTCTGGATTTCAGCAACAACCA-BBQ
	chicken	KF697090.1	
*TLR5*	turkey	HQ436463.1	F: AGCCTACTAGTGTGGCTAAATG R:ACACTGGTACACCTGCTAATG P: ROX-ACCAATGTAACCCTAGCTGGCTCA-BHQ2
	chicken	AJ626848.1	
*TLR6*	turkey	XM_019615148.1	F: CGAGCTGTACTTTGCCCATC R: GGTACCTCGCAGGGATAACA P: HEX-TGCTGGAGCCGATCCCTCCA-BHQ1
	chicken	NA ^#^	
*TLR7*	turkey	XM_010726471.2	F: CCAGATGCCTGCTATGATGC R: TCAGCTGAATGCTCTGGGAA P: FAM-TGGCTTCCAGGACAGCCAGTCT-BHQ1
	chicken	NM_001011688.2	
*TLR13*	turkey	XP_019475306.1	F-TGCTGGACCTGTCTCACAAT R: CAGGTTGCCCAAACTGTTGA P: ROX-CGGCTGACCACACTCGCCGA-BHQ2
	chicken	NA	
*TLR21*	turkey	NA	F: TCGCAACTGCATTGAGGATG R: ATGACAGATTGAGCGCGATG P: CY5-TTCCTGCAGTCGCCGGCCCT-BHQ2
	chicken	NM_001030558.1	

^#^ NA: not available; gene was not reported in NCBI database at the time of the establishment of the experiment.

**Table 2 vaccines-09-00960-t002:** Mean lesion scores of birds in every group at 4, 10 and 21 days post infection (DPI) for both species. Lesion score (LS) 0 represents no lesion, whereas LS 1 to 4 indicates mild to severe pathological changes.

DPI	Vaccinated Turkeys (VT)		Vaccinated Chickens (VC)		Vaccinated and Infected Turkeys (VIT)		Vaccinated and Infected Chickens (VIC)		Infected Turkeys (IT)		Infected Chickens (IC)	
	Caecum	Liver	Caecum	Liver	Caecum	liver	Caecum	Liver	Caecum	Liver	Caecum	Liver
**4**	0	0	0	0	2	0	1	0	2	1	2	1
**10**	0	1	0	1	3	1	1	0	4	4	3	3
**21**	2	0	1	0	4	3	0	1	NA ^#^	NA	1	1

^#^ NA: not applicable due to fatalities caused by the disease.

**Table 3 vaccines-09-00960-t003:** Overview on significant up- and down-regulations of TLR expressions due to vaccination and/or infection compared to control birds of that species in caecum and spleen. The colored fields represent significant variations. The number indicates at which day post inoculation significant changes occurred.

*Genes*	Species	Vaccinated		Vaccinated and Infected		Infected	
		Caecum	Spleen	Caecum	Spleen	Caecum	Spleen
*TLR1A*	turkey						
	chicken					21	
*TLR1B*	turkey	4		10		4, 10	
	chicken			4		10	
*TLR2A*	turkey						
	chicken						
*TLR2B*	turkey			4		4, 10	
	chicken			4		10	
*TLR3*	turkey		10			10	10
	chicken					4, 10	
*TLR4*	turkey			10, 21		4, 10	
	chicken	10		10		4	
*TLR5*	turkey		10				10
	chicken	10, 21		21		21	
*TLR6*	turkey		4	21			
	chicken	NA ^#^					
*TLR7*	turkey	10		10		10	
	chicken						
*TLR13*	turkey						
	chicken	NA					
*TLR21*	turkey	NA					
	chicken			4		4	
		significant up-regulation at specific DPI *p* ≤ 0.05		significant down-regulation at specific DPI *p* ≤ 0.05		non-significant changes at all three DPI *p* ≥ 0.05	

^#^ NA: not applicable due to unavailability of the relevant gene sequence.

## Data Availability

The data presented in this study are available on request from the corresponding author.

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
