# Peer review of "Vaccination against the Protozoan Parasite Histomonas meleagridis Primes the Activation of Toll-like Receptors in Turkeys and Chickens Determined by a Set of Newly Developed Multiplex RT-qPCRs"

_vaccines, 2021, doi:10.3390/vaccines9090960_

Round 1
Reviewer 1 Report
Reviewer Comments
The manuscript “Vaccination against the protozoan parasite Histomonas meleagridis primes the activation of toll-like receptors in turkeys and chickens determined by a set of newly developed multiplex RT-qPCRs” by Taniya Mitra and colleagues offers substantial proof to a longstanding question in the field of innate immune response to protozoan parasites in poultry birds. The authors have designed a RT-qPCR assay system to quantify TLR expression in H. meleagridis-infected turkeys and chickens to elegantly answer this question. The conclusions of this paper are well-supported with sufficient data and sound statistical analyses. However, there are some valid limitations to this manuscript which the authors need to address immediately. These are as follows:
- 1, lines 2-5 – The title of the manuscript could be more concise.
- 2, lines 59-61 – Could the authors shed light on the effect of intracellular protozoan parasites on the immune response of poultry? This could be of interest to the readers.
- 2 – The entire Methods section of this manuscript can be much better explained to the reader with a diagrammatic representation of the study protocol. The schematic representation would be in addition to the manuscript text.
- 4. Table 1 heading – Spelling of “Species” is wrong.
- 4, Materials and Methods Section: “Total RNA extraction and analysis for purity and integrity” is written ok but would do better with a reference. The following paper is apt for highlighting RNA extraction/RT-qPCR from animal samples:
Chattaraj, S., et al. Vitamin D as an important constituent of epididymal luminal micro-environment for maturation of spermatozoa in Large White Yorkshire boar. Vet. arhiv 89, 169-182, 2019. doi: 10.24099/vet.arhiv.0210.
- 5. Results lines 175-176 – “Non-vaccinated but infected turkeys (group IT) showed first clinical signs, such as depression, diarrhoea and ruffled feathers, starting on 7 DPI.”
Can the authors quantify these observed behavioral/physiological phenotypes in group IT turkeys?
- 6 and 8, Tables 2 and 3 – Please provide detailed legend for the tables so that readers can fully comprehend them.
- 9 and 10, Fig 1 – Poor graph representation in (A) and (B) is not allowing readers to fully grasp the interesting data. Individual bars are too small to appreciate significant differences between them. The asterisks (*) are all over the place in the figures. The authors may want to re-think data presentation (may use software like GraphPad) and also consider using professional figure-making software such as Adobe Illustrator/Photoshop. Figure legends need to be more descriptive also. The overall data representation needs to be vastly improved.
- 12 and 13, Fig 2 – Same comment as Fig. 1. The overall data representation needs to be vastly improved.
- 15 and 16, Fig 3 -This is better than the previous figures, but still need to be polished.
- 3 in Discussion, lines 371-380 – The authors may want to summarize their findings in the form of a graphical image/diagram to help the authors understand the conclusions better.
- Can the authors discuss how factors such as age, sex or diet of the poultry birds would affect innate immune response and TLR expression against meleagridis? Have the authors tested it?
Author Response
We appreciate the constructive comments and suggestions by the reviewer. Please find the attachment for the reply.

Reviewer 2 Report
Review - Manuscript ID: vaccines-1268763 “Vaccination against the protozoan parasite Histomonas meleagridis primes the activation of toll-like receptors in turkeys and chickens determined by a set of newly developed multiplex RT-qPCRs”
The authors examined the expression of various TLRs and pro-inflammatory cytokines in spleen and caecum samples from BUT 6 turkeys and VALO laying chickens in 4 experimental groups (vaccinated, infected, vaccinated and infected, and control) to evaluate the role of the innate immune response in Histomonas meleagridis infections.
In general, the study is within the scope of the journal, novel, and of interest to the poultry scientific community. However, some issues need to be addressed before a decision on publication can be made.
General comments
- The experiment was part of a larger study and some papers from it have already been published. These papers are frequently referenced in the M+M section. Although this is generally possible, it is not convenient for the reader to search for basic information in the cited papers. Please include relevant information on study design and methods in brief in the manuscript.
- Statistical analyses are described in a very rudimentary and incomprehensible manner. Which test assumptions were tested with which results? Fold changes of vaccinated and/or infected study groups were compared to control at each time point using t-tests. Why were not all study groups compared with each other? Was an attempt made to control for first-type error due to multiple testing?
- Only a simple univariate statistical analysis was performed. Please consider multivariable models to analyze the effect of possible independent variables (species, group, organ, time) on your study outcomes (expression changes).
- Results are not presented in an appropriate way (e.g., tables and figures are partially redundant). In descriptive statistics of TLR and cytokine expression, only parameters of location are given (mean), but no parameter of distribution (e.g., SD or CI). Results of multiplex RT-qPCR development are not shown in accordance to the MIQE guidelines.
Specific comments
Abstract: Please provide more information on study design (experimental groups, times of vaccination/infection and sample collection).
Keywords: Words from the title should be avoided.
Introduction: Please add your study hypotheses.
M+M, Trial setup: How was necessary sample size calculated? Which were your assumptions (major outcome, expected changes, alpha-error, statistical power)?
A timeline as an illustration to visualize the study design would be very helpful in terms of quick comprehensibility by the reader.
L95: Feed – Was this a commercial feed or an in-house diet? Please add specific designation and manufacturer or nutrient contents.
L. 104: Birds were previously determined for necropsy. Why no random selection at each time point? This may include a risk of bias – please discuss.
Please justify here why caecum and spleen were selected as organ samples. Why were liver samples not used additionally?
Please add a brief description of clinical scoring as well as pathological scoring of the organs.
Gene selection: Please justify cytokine selection at this stage.
Table 1: Spices = Species (header)? You used the same primers and probe sequences in both poultry species. May this affect results? Please discuss. Some TLR gene sequences for turkeys or chickens were not available in NCBI. Please explain in a footnote to the table how primers/probe were constructed and validated in this case.
L. 165: Which reference genes were used? All which are mentioned in the cited paper?
L. 167: Calculation of fold change: Please note the correct formatting (formula) and explain briefly the approach.
Statistical analysis: Did you check the assumptions of student´s t-test (for independent samples)? If normality of the data within each group is not given, non-parametric tests are more appropriate.
You performed a comparison of mean values (arithmetic means?) between controls and each group of interest (L. 171-172). Why were not all groups compared in an overall approach (e.g., one-factorial ANOVA within species with experimental group as fixed effect, and, if indicated, an appropriate post-hoc test)? In my opinion, it is also of interest to compare expression changes vaccinated vs. infected, vaccinated/infected vs. infected, and vaccinated/infected vs. vaccinated and not only the groups mentioned vs. control.
You calculated a plethora of t-tests which leads to an increase of alpha-error. Please implement an appropriate correction method (e.g., see https://www.aerzteblatt.de/int/archive/article/67552).
Did you analyse time-dependent changes within and between the experimental groups using statistical methods (i.e., interaction time*group)?
Did you analyze correlation between TLR and cytokine expression, and between expression patterns and clinical/pathological scores?
Results, Clinical signs and lesion scores: Clinical monitoring/scoring was not described in M+M. Please add.
Euthanasia: Please provide previously defined specific termination criteria.
Table 2: How was mean lesion score calculated? Please provide information in Stat. analysis section. Is the arithmetic mean an appropriate parameter of localization for scoring systems (i.e., ordinally scaled data)? I recommend calculating grouped medians and 95% CIs.
Tables should be self-explanatory. Please explain reason for NA in a footnote.
Data on validation of singleplex and multiplack RT-qPCRs are lacking completely. Please show essential information (qPCR validation and data analysis) according to the MIQE guidelines (Bustin et al., 2009; Table 1) in the manuscript or as supplemental material.
Table 3, Figures 1+2: Presentation of results appears hard to follow and information is partly redundant. Please think about a re-arrangemet.
Table 3: Header vaccinated – 2x caecum? NA – please explain.
Which numbers are given in the table cells? DPI? Please specify in table title.
This table is hard to read. Some table cells are underlined orange/green without a number (e.g., TLR 3 turkey – vaccinated caecum and TLR 5 turkey – vacc caecum), other cells contain numbers, but are not coloured (vaccinated – second column caecum)?
Figure 1: Asterisks appear dislocated/shifted. Was this a conversion problem? Please indicate that means of 3 birds are shown in the figure legends. Possibly, a table including fold changes (mean and SD) and corresponding p-values would be more informative.
Figure 2: see Fig. 1
Figure 3: Y-axis label is missing
Please indicate in the figure legends that data from infected turkey on 21 DPI is missing.
Discussion: L336-343 TLR expression correlated to clinical and pathological outcome – data not shown; local TLR expression was increased to a greater extend than the systemic immune response – was this statistically proven?
L. 349: cite by number
Please add a method discussion to outline strengths and limitations of your study. Furthermore, an outlook and specific suggestions for further studies should be added, including future relevance to the praxis and vaccine development.
Author Response
We appreciate the constructive comments and suggestions by the reviewer. Please find the attachment. Thank you.

Reviewer 3 Report
An interesting work, it develops primers and probes to test expression of toll like receptors in turkeys and chickens and proceeds to use them to study the innate immune response to infection with/out vaccination.
I would like to see more data on the validation of the testing protocol. Also, more could be discussed about the results. The whole submission could be shortened, and data need to be presented more precisely.
Abstract
Line 19-20, what is meant by ‘most significant? Are some variations less significant? Differences are either significant or not. Details needed (p value between specific differences)
Similarly, give details/values instead of ‘less prominent’ ‘earlier changes’
Can the conclusion of the abstract also be more specific? What is the insight?
Check italics
Line 43, to be effective
Methods
Should be shortened throughout, e.g. sentences like:
Line 73. Details on the animal trial are described in a previous work investigating changes of 73 leukocyte populations during histomonosi
Can be replaced with Animal trials were carried out as previously described (ref)
Line 125, what ratios were considered acceptable? Say it here and line 132-35 can be deleted
Line 174, refers to results obtained via approaches/analysis that have not been described and MandM
190 significantly downregulated compared to?
Mistake on table 3, vaccinated column has 2 identical sub-columns (caecum)
Fig 1 needs formatting, * do not match columns
Table 3, what do the number represent? Change fold? Details from legend missing
Legends for tables and figs need more details, including details of statistical analysis and of what * represents. E.g. * corresponds to p<=…. Between what values?
The statistical significance is indicated as * (p value ≤0.05).
Line 245, delete only
Iine 268 need references
Discussion
Needs to be more precise, better supported with literature, shorter and clearer; few examples of needed changes below:
Line 280-81 unclear
Line 300, what is a declining lesion? Smaller in size? In number?
Line 303, any support from literature for this speculation?
371, this is not the first time that the presence of an innate immune response in chicken has been revealed
How was the suitability of the primers confirmed? Any control? Size of bands of normal PCR, Sequencing of amplified fragments? How was the specificity of probes tested?
Author Response

(The authors gave the same response as above.)

Round 2
Reviewer 2 Report
Many thanks for the revised version. I can recommend publication in Vaccines.